# Adults' leisure-time physical activity preferences and association with physical activity guidelines by metropolitan status, United States, 2019

Christiaan G. Abildso[1,*], Eugene C. Fitzhugh[2☯], Alan M. Beck[3,‡], Ashleigh Johnson[4,‡], Dina L. Maruca[5,‡], Stefanie M. Meyer[6‡], Cynthia K. Perry[7‡], Carissa R. Smock[8,‡], Zachary Townsend[9‡], Lauren E. Jacobs[10‡], M. Renée Umstattd Meyer[11‡]

1 Division for Land-Grant Engagement, West Virginia University Extension, West Virginia University, Morgantown, West Virginia, United States of America, 2 Department of Kinesiology, Recreation, and Sports Studies; University of Tennessee at Knoxville; Knoxville, Tennessee, United States of America, 3 School of Public Health, Washington University in St. Louis, St. Louis, Missouri, United States of America, 4 School of Exercise and Nutritional Sciences, San Diego State University, San Diego, California, United States of America, 5 Department of Orthopaedics and Division of Physical Therapy, West Virginia University, Morgantown, West Virginia, United States of America, 6 School of Health Professions, Concordia College, Moorhead, Minnesota, United States of America, 7 School of Nursing, Oregon Health & Science University, Portland, Oregon, United States of America, 8 School of Business and Economics, National University, San Diego, California, United States of America, 9 School of Health Sciences, Salisbury University, Salisbury, Maryland, United States of America, 10 School of Kinesiology, Physical Education, and Athletic Training; College of Education and Human Development; University of Maine, Orono, Maine, United States of America, 11 Robbins College of Health and Human Sciences, Baylor University, Waco, Texas, United States of America

☯ These authors contributed equally to this work.
‡ AMB, AJ, DLM, SMM, CKP, CRS, ZT, LEJ, MRUM authors contributed equally to this work.
* cgabildso@mail.wvu.edu

## Abstract

### Background

Rural US adults experience disparities in meeting the physical activity guidelines (PAGs). Little is known about preferred types of leisure-time physical activity (LTPA), nor their relationship with meeting the PAGs. This study aimed to identify the most prevalent LTPAs among US adults, the relationship between LTPA types and meeting PAGs, and how these differed by residential status.

### Methods

2019 Behavioral Risk Factor Surveillance Study data were analyzed. Age-adjusted prevalence of 75 LTPAs and 11 categories of LTPAs that respondents spent the "most time" and "next most time" in during the previous month were compared by residential status (metropolitan; non-metropolitan). Logistic and multinomial logistic regression analyses adjusted for a variety of factors were used to compare the prevalence of meeting aerobic-, muscle strengthening, and combined PAGs by residential status. Age-adjusted prevalence of meeting the combined PAGs by 11 LTPA categories was compared by metropolitan status.

**Data availability statement:** The data underlying the results presented in the study are available from the National Center for Chronic Disease Prevention and Health Promotion, Division of Population Health at the Centers for Disease Control and Prevention: https://www.cdc.gov/brfss/annual_data/annual_2019.html.

**Funding:** This project was supported by cooperative agreement number U48DP006381 from the Centers for Disease Control and Prevention's (CDC) Division of Nutrition, Physical Activity, and Obesity (DNPAO). The work presented is the responsibility of the authors and does not necessarily represent the official position of CDC, the Department of Health and Human Services or the federal Government.

**Competing interests:** I have read the journal's policy and the authors of this manuscript report receiving no financial support for this work or any competing interests. However, Dr. Johnson was supported by a Research Career Development Award from the National Heart, Lung, and Blood Institute (K01HL171860-01) during the period in which the research took place.

## Results

Walking was the most prevalent LTPA (44.1%). Lawn/garden, hunting/fishing, household, farm/ranch work, childcare, and winter activities were more prevalent among non-metropolitan (rural) residents. Walking, running/jogging, weightlifting, conditioning, other activities, sports, bicycling, water activities, and dance were more prevalent among metropolitan (urban) residents. Non-metropolitan residents were less likely to meet the minimal aerobic-, muscle-strengthening, and combined PAGs, and were more likely to be inactive. Among adults that engaged in walking, roughly 25% met the combined PAGs and about 22% did not meet either PAG.

## Conclusions

When creating targeted rural PA interventions, LTPA preferences could be embraced and augmented or the non-preferred LTPAs could be adapted.

---

## Introduction

Increasing the prevalence of meeting the physical activity guidelines (PAGs) is vital for public health [1]. Despite population-level increases in meeting the aerobic, muscle strengthening, and combined PAGs, and corresponding decreases in inactivity over the last quarter century among adults in the United States (US) [2,3], there remain pernicious disparities by sociodemographic characteristics, region, sex, and urbanicity/rurality [3–7].

Population-level physical activity (PA) promotion should be targeted to the activities people prefer [8], what they have – or could have – access to [9], and what is culturally appropriate [10–14], safe [15], enjoyable [16]. and effective (i.e., likelihood the activity will help people meet PAGs) [17,18]. Similar to the frequency, intensity, time, and type (FITT) principle of individual-level exercise prescription [19]. population surveillance documenting the prevalence of meeting PAGs (i.e., the frequency, intensity, time) would benefit from greater understanding of the type of activities people engage in to meet PAGs. This information would provide valuable insight into what is popular for population subgroups and what types of activities are associated with meeting the PAGs. For example, people who cycle during their leisure time are more likely to meet PAGs, yet rural adults are less likely to cycle [20]. Thus, before promoting cycling to meet PAGs in a rural setting, factors such as the expense of owning a bicycle, and lack of a supportive physical and cultural environment may need to be addressed.

It is also particularly important to target population-level PA promotion in areas and among demographic groups with persistently low rates of leisure-time physical activity (LTPA) [1]. In the US, rural-residing adults have lagged behind adults in urban areas in the prevalence of meeting the aerobic, muscle strengthening, and combined PAGs for decades [4,5,7]. Specific to rural areas, the prevalence of meeting the combined PAGs is lower among women, older adults, individuals who did not attend college, individuals of Hispanic ethnicity, and individuals living in the US South [4].

Little is known about US adults' preferred types of LTPA [8], especially those residing in rural areas. The most recent research on LTPA type preferences, using 2011 Behavior Risk Factor Surveillance System (BRFSS) data, showed walking was the most popular LTPA among US adults, with differences by age and sex [8]. Women were more likely to engage in walking than men (especially those under 65 years old) and dancing/aerobics activities across all ages; men were more likely to engage in higher intensity activities (i.e., sports, bicycling, running/jogging) across all age groups [8]. These results are concordant with analyses of NHANES data from the 2003−04 and 2005−06 data collection cycles, which found walking was by far the most popular LTPA, though engaged in at a level, on average, below what is necessary to meet PAGs [17]. Importantly, this body of literature has not been updated since the revision of the PAGs in 2018; nor has it included muscle strengthening activities. To address these needs, the objective of this study was to identify rural/urban differences in the prevalence of specific types of LTPA, the prevalence of meeting PAGs, and the relationship between LTPA types and meeting PAGs among US adults.

## Methods

This study utilized de-identified data from the 2019 BRFSS public-use dataset, downloaded on August 20, 2024. The BRFSS is a collaborative project of the Centers for Disease Control and Prevention (CDC) with the 50 states, the District of Columbia, and 3 US territories. Utilizing telephone surveys of the noninstitutionalized adult population (18 years and older) within the US, the focus of the BRFSS is to measure health-related risk behaviors, chronic health conditions, and preventive services [21].

### Sample

The 2019 BRFSS contained data from 418,268 adults. Of these, the final sample for this study consisted of 396,261 adults who completed the PA core section of the survey.

### Demographic characteristics

Analyses included the following demographic characteristics that are associated with PA: sex (male, female), age group (18–29, 30–39, 40–49, 50–59, 60–69, 70–79, or 80+years), race/ethnicity (non-Hispanic white, non-Hispanic black, non-Hispanic other race, non-Hispanic multiracial, or Hispanic), education (less than high school, high school, some college, or college graduate), household income (less than $15,000, $15,000 to less than $24,999, $25,000 to less than $35,000, $35,000 to less than $50,000, or $50,000 or more), perceived health (excellent, very good, good, fair, or poor), and body mass index (BMI) based on self-reported height and weight (underweight, normal weight, overweight, or obese). Additional variables associated with PA were included in the analyses, specifically the US Census region of the respondent household (Northeast, South, Midwest, or West) [22] and the interview month for each respondent (January through December).

### Physical activity measures

In the PA core section of the 2019 BRFSS, LTPA was assessed by asking respondents if they participated "in any physical activities or exercises such as running, calisthenics, golf, gardening, or walking for exercise" over the past month that were not part of their regular job. If responding yes, they were then prompted to report the LTPA they spent the "most time" and, separately, the "next most time" engaging in during the previous month. Follow-up questions were used to ascertain the frequency of these activities per week or month, and the average amount of time they spent when doing a bout of the activity. Respondents could report participating in 74 unique physical activities. These were consolidated into 11 categories for analyses by classifying the activities into the 9 general categories developed by Watson and colleagues [8] based on the Compendium of Physical Activities: [23] conditioning, dance, hunting/fishing, lawn/garden, running/jogging, sports, walking, water activities, and winter activities. Weightlifting, which was excluded from the conditioning category by Watson

et al., [8] was included in our analyses as its own category rather than a conditioning activity as the Compendium [23] does, to isolate muscular strengthening activities for analysis with muscle strengthening physical activity guidelines and to maintain comparability of the Conditioning category with Watson et al., [8] which was only focused on aerobic LTPA analyses. "Other activity," which respondents could report if they engaged in a LTPA not on the list of 74, was also included as its own category, for a total of 75 categories.

The BRFSS also provided several calculated measures used to classify respondents as meeting the aerobic, muscle-strengthening, and combined PAGs [1,24].

**Aerobic physical activity.** Each type of aerobic PA reported by a respondent is classified in the BRFSS as moderate- or vigorous intensity based on the Compendium of Physical Activities [23] and used to calculate the total minutes of each intensity level per week. The minutes engaged in moderate-intensity aerobic LTPA (MPA) per week and the equivalent minutes of vigorous-intensity aerobic LTPA (VPA; 1 minute of VPA = 2 minutes of MPA) per week were used to classify respondents in 2 ways. First, for a dichotomous "minimal aerobic" PAG variable as either meeting (≥150 minutes of moderate-to-vigorous-intensity aerobic activity; MVPA) or not meeting (<150 minutes of MVPA) the minimal aerobic PAG. Secondly, for a 4-level "high aerobic" PAG variable as: highly active (≥300 minutes of MVPA), active (150–299 minutes of MVPA), insufficiently active (1–149 minutes MVPA), or inactive (0 minutes MVPA).

**Muscle strengthening physical activity.** Respondents met the muscle-strengthening activity guideline if they reported 2 or more days of muscle-strengthening activities per week. Alternatively, not meeting the muscle-strengthening activity guideline was determined if they reported less than 2 days of muscle-strengthening activities per week.

**Combined aerobic and muscle strengthening physical activity.** Finally, respondents were classified into 1 of 4 categories for meeting the combined aerobic and muscle-strengthening PAG: (1) met both PAGs (≥150 minutes of aerobic LTPA and ≥2 days of muscle-strengthening activity per week), (2) met aerobic PAG only (≥150 minutes of aerobic LTPA and <2 days of muscle-strengthening activity per week), (3) met muscle-strengthening activity guideline only (<150 minutes of aerobic LTPA and ≥2 days of muscle-strengthening activity per week), (4) did not meet either PAG (<150 minutes of aerobic LTPA and <2 days of muscle-strengthening activity per week).

## Metropolitan status

The 2013 National Center for Health Statistics (NCHS) Urban/Rural Classification Scheme [25] was used as a dichotomous measure of urbanization of each respondent's county of residence in the 2019 BRFSS database [23]. This measure, labeled _METSTAT in the BRFSS data dictionary, identifies a metropolitan (i.e., urban) county, including the associated metropolitan statistical area (MSA), as being linked to 4 of the 6 categories of the NCHS Urban/Rural Scheme – large central metro (county in an MSA of ≥1 million population that [a] contains the entire population of the largest principal city of the MSA, [b] is completely contained within the largest principal city of the MSA, or [c] contains at least 250,000 residents of any principal city in the MSA), large fringe metro (county in an MSA of ≥1 million population that doesn't qualify as large central), medium metro (county in MSA of 250,000–999,999 population), and small metro (county in MSA with <250,000 population) [25]. Respondents were classified as living in a non-metropolitan (i.e., rural) area if their county of residence was linked to the final two levels of urbanization -- micropolitan (counties with urban cluster population of 10,000–49,000) and 'noncore' counties which includes non-metropolitan counties not classified as micropolitan [25].

## Statistical analyses

After the initial phase of variable recoding using SAS 9.4 [26], SAS-callable SUDAAN [27] was used to adjust for the BRFSS complex sampling design for all analyses, including examining all related assumptions. Prevalence estimates of physical activities were age-adjusted to the 2010 US adult population. The SAS procedure PROC DESCRIPT was used to determine the age-adjusted prevalence estimates and 95% Confidence Intervals (CIs), and for metropolitan and non-metropolitan comparisons of prevalence rates of (1) engaging in individual LTPA types and (2) engaging in categories of

LTPAs. Due to multiple tests in the analyses specific to determining differences in prevalence estimates between metropolitan and non-metropolitan counties of residence in the US, the Bonferroni method was used to adjust for alpha inflation. Specifically, when comparing the 75 individual LTPA types by metropolitan status, the p-value was set at $p = 0.0007$ (=0.05/75). For comparisons across the 11 LTPA categories the p-value was set at $p = 0.0045$ (=0.05/11). Age-adjusted prevalence of meeting the combined PAGs for each of the 11 LTPA categories was compared between non-metropolitan and metropolitan (referent) and residents using SAS procedure PROC DESCRIPT.

The SAS procedure PROC RLOGIST was used to estimate the adjusted odds ratios (AORs) of meeting the minimal aerobic PAG and muscle-strengthening activity guideline by comparing non-metropolitan residents against metropolitan (referent) when adjusted for age, region, sex, race/ethnicity, education, income, month of survey response, self-reported health, and BMI. Similarly, for the 4-level high aerobic PAG and the 4-level combined PAG outcomes analyses, SAS procedure PROC MULTILOG was used to generate predicted marginal proportions of the outcomes and model-adjusted relative risk ratios (RRRs) among non-metropolitan residents relative to metropolitan residents (referent).

## Results

Information about the 2019 BRFSS participants and population-weighted analytical comparisons of metropolitan and non-metropolitan participants are displayed in Table 1. Analyses revealed significant demographic differences between metropolitan and non-metropolitan populations. Non-metropolitan populations tended to be older, less racially/ethnically diverse, less likely to have earned a 4-year college degree, have a lower income, have fair or poor perceived health, and have obesity (all $p < .0001$).

### LTPA Types by metropolitan status

The age-adjusted prevalence of all LTPA types, categories of LTPA types, and comparisons by metropolitan status are shown in Table 2 (categories ordered from highest to lowest prevalence). The most popular LTPAs were walking (44.1%, 95% CI = 43.8-44.4%), running (11.7%, 95% CI = 11.5-11.9%), weightlifting (10.7%, 95% CI = 10.4-10.9%), gardening (5.8%, 95% CI = 5.7-6.0%), and bicycling (3.8%, 95% CI = 3.7-4.0%). Of these, only gardening was more prevalent among residents of non-metropolitan (6.9%) than metropolitan (5.7%) areas; the others were more prevalent among metropolitan residents. Nearly one-tenth of respondents reported "other activity" (9.3%; 95% CI = 9.1-9.5%).

Table 2 also displays the metropolitan and non-metropolitan comparison of LTPAs when condensed to the 11 activity categories. Differences demonstrated by these analyses include significantly higher participation in lawn/garden, hunting/fishing, and winter activities in non-metropolitan residents than metropolitan residents; and significantly higher participation in walking, running/jogging, weightlifting, conditioning, sports, dance, and water activities by metropolitan residents.

### Prevalence of meeting PAGs by metropolitan status

Metropolitan and non-metropolitan residents differed in the prevalence of meeting the aerobic and muscle strengthening PAGs. In the fully adjusted models displayed in Table 3, residents of non-metropolitan counties were 6% less likely to meet the aerobic PAG (AOR = 0.94; 95% CI 0.91-0.97) and 8% less likely to meet the muscle-strengthening activity guideline (AOR = 0.92; 95% CI 0.89-0.96) than metropolitan residents.

Metropolitan and non-metropolitan residents also differed in the prevalence of meeting the 4-level high aerobic- and combined PAG measures. Non-metropolitan residents were 11% more likely to be inactive than metropolitan residents (29.9% versus 26.9%, respectively) and 3% more likely to meet the highly active PAG than metropolitan residents (33.9% versus 32.8%, respectively). Non-metropolitan residents were 5% less likely to meet the combined PAGs than metropolitan residents (22.6% versus 23.9%, respectively) and 5% more likely to meet neither PAG than metropolitan residents (37.2% versus 35.3%, respectively).

**Table 1. Selected Participant Characteristics and Population-weighted Comparisons by Metropolitan Status: 2019 BRFSS.**

| | Total | | Metro | | Non-Metro | | |
|---|---|---|---|---|---|---|---|
| | n | % (SE) | n | % (SE) | n | % (SE) | p |
| **Gender** | | | | | | | 0.986 |
| Male | 186,445 | 48.7 (0.2) | 129,658 | 48.7 (0.2) | 56,787 | 48.7 (0.2) | |
| Female | 223,365 | 51.3 (0.2) | 152,994 | 51.3 (0.2) | 70,371 | 51.3 (0.2) | |
| **Age, years** | | | | | | | <0.001 |
| 18-29 | 44,647 | 20.3 (0.1) | 34,042 | 20.7 (0.2) | 10,605 | 17.8 (0.3) | |
| 30-39 | 46,713 | 17.9 (0.1) | 34,191 | 17.9 (0.2) | 12,522 | 15.2 (0.2) | |
| 40-49 | 50,448 | 15.4 (0.1) | 36,080 | 15.7 (0.1) | 14,368 | 14.1 (0.2) | |
| 50-59 | 71,541 | 16.8 (0.1) | 49,041 | 16.7 (0.1) | 22,500 | 17.4 (0.2) | |
| 60-69 | 90,405 | 15.4 (0.1) | 59,396 | 14.9 (0.1) | 31,009 | 18.1 (0.2) | |
| 70-79 | 70,581 | 10.1 (0.1) | 46,560 | 9.7 (0.1) | 24,021 | 12.0 (0.2) | |
| 80+ | 35,475 | 4.6 (0.1) | 23,342 | 4.4 (0.1) | 12,133 | 5.4 (0.1) | |
| **Race/Ethnicity** | | | | | | | <0.001 |
| White NH | 310,434 | 62.6 (0.2) | 205,119 | 59.4 (0.2) | 105,315 | 80.1 (0.3) | |
| Black NH | 31,391 | 12.0 (0.1) | 25,842 | 12.8 (0.1) | 5,549 | 7.5 (0.2) | |
| Other Race NH | 19,631 | 7.3 (0.1) | 13,762 | 8.0 (0.1) | 5,869 | 3.6 (0.1) | |
| Multiracial NH | 8,179 | 1.3 (0.0) | 5,684 | 1.3 (0.0) | 2,495 | 1.4 (0.1) | |
| Hispanic | 31,277 | 16.8 (0.1) | 25,782 | 18.6 (0.2) | 5,495 | 7.4 (0.2) | |
| **Education** | | | | | | | <0.001 |
| <High School | 28,939 | 12.8 (0.1) | 18,934 | 12.5 (0.1) | 10,005 | 14.6 (0.2) | |
| High School | 109,444 | 27.8 (0.1) | 68,577 | 28.4 (0.2) | 40,867 | 35.4 (0.3) | |
| Some College | 114,480 | 31.0 (0.2) | 76,948 | 30.8 (0.2) | 37,532 | 32.0 (0.3) | |
| College+ | 155,139 | 28.4 (0.1) | 116,850 | 30.2 (0.2) | 38,289 | 17.9 (0.2) | |
| **Income** | | | | | | | <0.001 |
| <$15,000 | 29,659 | 9.8 (0.1) | 18,785 | 9.5 (0.1) | 10,874 | 11.3 (0.2) | |
| $15,000 to <$25,000 | 51,415 | 15.6 (0.1) | 32,959 | 15.1 (0.2) | 18,456 | 18.7 (0.2) | |
| $25,000 to <$35,000 | 33,701 | 9.8 (0.1) | 21,822 | 9.5 (0.1) | 11,879 | 11.7 (0.2) | |
| $35,000 to <$50,000 | 45,927 | 12.8 (0.1) | 30,133 | 12.4 (0.1) | 15,794 | 14.9 (0.2) | |
| ≥$50,000 | 170,758 | 52.1 (0.2) | 124,997 | 53.6 (0.2) | 45,761 | 43.4 (0.3) | |
| **Perceived Health** | | | | | | | <0.001 |
| Excellent | 64,519 | 17.4 (0.1) | 46,890 | 17.9 (0.1) | 17,629 | 14.2 (0.2) | |
| Very Good | 134,353 | 31.6 (0.2) | 94,378 | 31.9 (0.2) | 39,975 | 30.3 (0.3) | |
| Good | 130,803 | 32.4 (0.2) | 89,184 | 32.2 (0.2) | 41,619 | 33.2 (0.3) | |
| Fair | 57,432 | 13.9 (0.1) | 38,094 | 13.6 (0.1) | 19,338 | 15.8 (0.2) | |
| Poor | 21,677 | 4.7 (0.1) | 13,425 | 4.4 (0.1) | 8,252 | 6.6 (0.1) | |
| **Body Mass Index** | | | | | | | <0.001 |
| Underweight | 6,434 | 1.9 (0.1) | 4,498 | 1.9 (0.1) | 1,936 | 1.9 (0.1) | |
| Normal weight | 113,594 | 31.5 (0.2) | 80,720 | 32.3 (0.2) | 32,874 | 27.3 (0.3) | |
| Overweight | 133,442 | 35.2 (0.2) | 92,015 | 35.4 (0.2) | 41,427 | 34.1 (0.3) | |
| Obese | 120,603 | 31.4 (0.2) | 79,900 | 30.4 (0.2) | 40,703 | 36.8 (0.3) | |

Footnotes: Percentage and Standard Error are population-weighted analytical comparisons of metropolitan and non-metropolitan residents

NH = non-Hispanic; Body Mass Index (BMI): Underweight, BMI < 18.5 kg/m$^2$; Normal weight, BMI 18.5–24.9 kg/m$^2$; Overweight, BMI 25–29.9 kg/m$^2$; Obese, BMI ≥ 30 kg/m$^2$

**Table 2. Age-Adjusted Rates of Adults Participating in Specific Leisure-Time Physical Activities by Metropolitan Status – 2019 BRFSS.**

| Leisure-Time Physical Activities by Category (underlined) | All % | (95% CI) | Metropolitan % | (95% CI) | Non-Metropolitan % | (95% CI) |
|---|---|---|---|---|---|---|
| **Walking (Total)** | **45.6** | **(45.3-45.9)** | **45.9** | **(45.5-46.2)** | **44.6** | **(44.0-45.2)** |
| Backpacking[a] | 0.03 | (0.02-0.03) | 0.03 | (0.02-0.04) | 0.02 | (0.01-0.03) |
| Hiking – cross-country[a] | 1.99 | (1.90-2.08) | 1.98 | (1.88-2.09) | 2.01 | (1.85-2.18) |
| Walking | 44.12 | (43.79-44.44) | 44.38 | (44.01-44.75) | 43.03 | (42.41-43.64) |
| **Running/Jogging (Total)** | **13.1** | **(12.8-13.3)** | **13.6** | **(13.4-13.9)** | **9.8** | **(9.4-10.3)** |
| Jogging | 1.41 | (1.31-1.51) | 1.48 | (1.37-1.60) | 0.99 | (0.86-1.12) |
| Running | 11.70 | (11.47-11.93) | 12.17 | (11.91-12.43) | 8.87 | (8.46-9.28) |
| **Weightlifting** | **10.7** | **(10.4-10.9)** | **11.2** | **(10.9-11.4)** | **7.8** | **(7.4-8.2)** |
| **Conditioning (Total)** | **9.5** | **(9.3-9.7)** | **10.0** | **(9.8-10.2)** | **6.3** | **(6.0-6.6)** |
| Active gaming devices (e.g., Wii Fit, Dance Revolution)[a] | 0.06 | (0.04-0.08) | 0.06 | (0.04-0.08) | 0.04 | (0.02-0.06) |
| Bicycling machine exercise | 2.19 | (2.10-2.29) | 2.33 | (2.22-2.43) | 1.46 | (1.34-1.59) |
| Stair climbing/Stair master | 0.59 | (0.54-0.64) | 0.61 | (0.56-0.66) | 0.49 | (0.41-0.56) |
| Calisthenics | 2.47 | (2.36-2.57) | 2.60 | (2.47-2.72) | 1.70 | (1.52-1.88) |
| Elliptical/EFX machine exercise | 1.63 | (1.55-1.72) | 1.73 | (1.63-1.83) | 1.12 | (0.99-1.24) |
| Rowing machine exercise | 0.20 | (0.17-0.23) | 0.21 | (0.18-0.25) | 0.12 | (0.08-0.16) |
| Wrestling[a] | 0.08 | (0.06-0.10) | 0.08 | (0.06-0.10) | 0.06 | (0.03-0.08) |
| Yoga | 2.25 | (2.15-2.35) | 2.40 | (2.29-2.52) | 1.36 | (1.25-1.48) |
| Pilates | 0.31 | (0.28-0.35) | 0.35 | (0.30-0.39) | 0.15 | (0.10-0.19) |
| **Other Activities** | **9.3** | **(9.1-9.5)** | **9.4** | **(9.2-9.7)** | **8.3** | **(8.0-8.7)** |
| **Lawn/Garden (Total)** | **8.6** | **(8.4-8.8)** | **8.2** | **(8.0-8.4)** | **10.8** | **(10.5-11.2)** |
| Gardening (e.g., spading, weeding, digging, filling) | 5.84 | (5.70-5.98) | 5.67 | (5.51-5.82) | 6.86 | (6.56-7.16) |
| Yard work (e.g., cut/gather wood, trimming hedges) | 2.83 | (2.74-2.92) | 2.59 | (2.49-2.68) | 4.15 | (3.93-4.37) |
| Raking lawn | 0.05 | (0.04-0.06) | 0.04 | (0.03-0.05) | 0.08 | (0.05-0.11) |
| Mowing lawn | 0.47 | (0.43-0.51) | 0.40 | (0.36-0.44) | 0.88 | (0.72-1.03) |
| **Sports (Total)** | **6.5** | **(6.3-6.7)** | **6.7** | **(6.5-6.9)** | **5.3** | **(5.0-5.6)** |
| Basketball[a] | 1.42 | (1.33-1.50) | 1.44 | (1.34-1.54) | 1.29 | (1.12-1.47) |
| Badminton[a] | 0.04 | (0.03-0.06) | 0.04 | (0.03-0.06) | 0.02 | (0.00-0.05) |
| Bowling[a] | 0.21 | (0.19-0.24) | 0.21 | (0.19-0.24) | 0.22 | (0.15-0.29) |
| Frisbee[a] | 0.07 | (0.06-0.09) | 0.07 | (0.05-0.09) | 0.08 | (0.04-0.13) |
| Golf (with motorized cart)[a] | 1.20 | (1.14-1.26) | 1.20 | (1.13-1.26) | 1.26 | (1.14-1.39) |
| Golf (without motorized cart) | 0.60 | (0.55-0.65) | 0.63 | (0.59-0.59) | 0.42 | (0.36-0.48) |
| Handball[a] | 0.01 | (0.01-0.02) | 0.01 | (0.01-0.02) | 0.01 | (0.00-0.01) |
| Hockey | 0.13 | (0.11-0.15) | 0.14 | (0.11-0.15) | 0.08 | (0.05-0.10) |
| Horseback riding | 0.20 | (0.17-0.24) | 0.18 | (0.15-0.22) | 0.34 | (0.26-0.41) |
| Inline skating[a] | 0.01 | (0.00-0.02) | 0.01 | (0.00-0.02) | 0.01 | (0.00-0.01) |
| Karate/martial Arts | 0.26 | (0.22-0.30) | 0.27 | (0.23-0.32) | 0.16 | (0.10-0.22) |
| Lacrosse | 0.04 | (0.02-0.05) | 0.04 | (0.02-0.06) | 0.01 | (0.00-0.02) |
| Mountain climbing[a] | 0.03 | (0.02-0.03) | 0.02 | (0.01-0.03) | 0.04 | (0.01-0.06) |
| Paddleball[a] | 0.02 | (0.01-0.02) | 0.02 | (0.01-0.02) | 0.01 | (0.00-0.02) |
| Racquetball | 0.10 | (0.08-0.11) | 0.10 | (0.08-0.13) | 0.06 | (0.03-0.08) |
| Rock climbing | 0.18 | (0.15-0.20) | 0.19 | (0.16-0.22) | 0.10 | (0.05-0.16) |
| Rope skipping | 0.04 | (0.02-0.05) | 0.04 | (0.02-0.06) | 0.01 | (0.00-0.02) |
| Rugby[a] | 0.03 | (0.02-0.04) | 0.03 | (0.02-0.06) | 0.01 | (0.00-0.03) |
| Skateboarding[a] | 0.16 | (0.13-0.18) | 0.16 | (0.13-0.19) | 0.11 | (0.06-0.16) |
| Soccer | 0.84 | (0.76-0.91) | 0.92 | (0.83-1.00) | 0.36 | (0.27-0.45) |

*(Continued)*

**Table 2.** (Continued)

| | All | | Metropolitan | | Non-Metropolitan | |
|---|---|---|---|---|---|---|
| Softball/Baseball[a] | 0.28 | (0.25-0.31) | 0.28 | (0.24-0.31) | 0.31 | (0.25-0.38) |
| Squash[a] | 0.02 | (0.01-0.03) | 0.02 | (0.01-0.03) | 0.01 | (0.00-0.02) |
| Table tennis[a] | 0.02 | (0.01-0.03) | 0.02 | (0.01-0.03) | 0.01 | (0.00-0.02) |
| Tai Chi | 0.08 | (0.07-0.09) | 0.09 | (0.07-0.10) | 0.06 | (0.04-0.08) |
| Tennis | 0.50 | (0.45-0.56) | 0.55 | (0.49-0.61) | 0.24 | (0.18-0.29) |
| Touch football[a] | 0.04 | (0.02-0.05) | 0.03 | (0.02-0.05) | 0.05 | (0.02-0.08) |
| Volleyball[a] | 0.30 | (0.26-0.35) | 0.31 | (0.26-0.36) | 0.27 | (0.20-0.34) |
| **Water Activities (Total)** | **2.9** | **(2.8-3.0)** | **3.0** | **(2.9-3.2)** | **2.0** | **(1.8-2.2)** |
| Boating (e.g., canoeing, rowing, kayaking, sailing)[a] | 0.12 | (0.10-0.14) | 0.12 | (0.09-0.14) | 0.13 | (0.09-0.17) |
| Canoeing/rowing in competition[a] | 0.06 | (0.05-0.07) | 0.06 | (0.04-0.07) | 0.07 | (0.05-0.09) |
| Scuba diving[a] | 0.02 | (0.01-0.02) | 0.02 | (0.01-0.02) | 0.01 | (0.00-0.02) |
| Snorkeling[a] | 0.01 | (0.00-0.02) | 0.01 | (0.00-0.02) | 0.00 | (0.00-0.01) |
| Surfing | 0.07 | (0.05-0.09) | 0.08 | (0.06-0.10) | 0.03 | (0.02-0.05) |
| Swimming | 2.01 | (1.91-2.10) | 2.10 | (1.98-2.21) | 1.56 | (1.40-1.71) |
| Swimming in laps | 0.62 | (0.57-0.67) | 0.68 | (0.63-0.74) | 0.27 | (0.22-0.32) |
| Waterskiing[a] | 0.04 | (0.03-0.05) | 0.04 | (0.03-0.05) | 0.04 | (0.02-0.06) |
| **Dance** | **2.7** | **(2.6-2.8)** | **2.9** | **(2.8-3.0)** | **1.9** | **(1.7-2.0)** |
| Aerobics video or class | 1.66 | (1.58-1.74) | 1.73 | (1.64-1.82) | 1.29 | (1.16-1.41) |
| Dancing (e.g., ballet, ballroom, Latin, hip hop) | 1.09 | (1.01-1.17) | 1.17 | (1.08-1.26) | 0.59 | (0.47-0.71) |
| **Fishing/Hunting (Total)** | **0.4** | **(0.37-0.45)** | **0.3** | **(0.3-0.4)** | **0.8** | **(0.7-1.0)** |
| Fishing from riverbank or boat | 0.24 | (0.21-0.27) | 0.20 | (0.17-0.23) | 0.46 | (0.36-0.56) |
| Hunting large game – deer, elk | 0.12 | (0.10-0.14) | 0.08 | (0.06-0.11) | 0.31 | (0.25-0.37) |
| Hunting small game – quail | 0.06 | (0.04-0.08) | 0.06 | (0.03-0.08) | 0.10 | (0.07-0.12) |
| Stream fishing in waders[a] | 2.00 | (2.00-2.01) | 2.00 | (2.00-2.01) | 2.00 | (2.00-2.00) |
| **Winter Activities (Total)** | **0.4** | **(0.4-0.5)** | **0.4** | **(0.35-0.41)** | **0.7** | **(0.6-0.8)** |
| Sledding, tobogganing[a] | 0.00 | (0.00-0.01) | 0.00 | (0.00-0.01) | 0.01 | (0.00-0.01) |
| Skating – ice or roller | 0.07 | (0.06-0.09) | 0.08 | (0.06-0.09) | 0.04 | (0.02-0.06) |
| Snow blowing | 0.02 | (0.01-0.02) | 0.02 | (0.01-0.02) | 0.04 | (0.02-0.05) |
| Snow shoveling by hand | 0.20 | (0.18-0.22) | 0.16 | (0.14-0.18) | 0.40 | (0.35-0.46) |
| Snow skiing | 0.13 | (0.11-0.15) | 0.12 | (0.10-0.14) | 0.20 | (0.16-0.24) |
| Snowshoeing | 0.02 | (0.01-0.02) | 0.01 | (0.01-0.02) | 0.06 | (0.03-0.09) |
| **Miscellaneous Activities** | | | | | | |
| Bicycling | 3.81 | (3.68-3.95) | 4.02 | (3.87-4.17) | 2.68 | (2.51-2.86) |
| Boxing | 0.26 | (0.23-0.30) | 0.28 | (0.24-0.32) | 0.15 | (0.09-0.20) |
| Carpentry[a] | 0.13 | (0.11-0.15) | 0.12 | (0.09-0.14) | 0.20 | (0.15-0.24) |
| Childcare | 0.72 | (0.66-0.78) | 0.68 | (0.61-0.75) | 0.99 | (0.86-1.12) |
| Farm/Ranch work (e.g., care for livestock, stack hay) | 0.50 | (0.46-0.53) | 0.32 | (0.28-0.35) | 1.46 | (1.33-1.60) |
| Household activities (e.g., vacuum, dust, home repair) | 2.56 | (2.47-2.65) | 2.44 | (2.34-2.54) | 3.20 | (3.00-3.39) |
| Painting/papering house[a] | 0.05 | (0.04-0.06) | 0.05 | (0.04-0.06) | 0.07 | (0.04-0.10) |
| Upper body cycle (e.g., wheelchair sports, ergometer)[a] | 0.11 | (0.09-0.13) | 0.11 | (0.09-0.13) | 0.10 | (0.07-0.14) |

Note: [a]Activity is not significantly different between metropolitan and non-metropolitan counties.

EFX = Elliptical Fitness Crosstrainer

**Table 3. Adjusted and Weighted Proportion\* of Adults Meeting the Leisure-Time Aerobic, Muscle Strengthening, and Combined Physical Activity Guidelines by Metropolitan Residential Status – 2019 BRFSS.**

**High Aerobic (4-level)**

| | Metropolitan, % (95% CI) (ref) | Non-Metropolitan, % (95% CI) | RRR | 95% CI |
|---|---|---|---|---|
| Highly Active | 32.8 (32.4-33.2) | 33.9 (33.2-34.6) | 1.03 | 1.01-1.06 |
| Active | 19.2 (18.9-19.6) | 16.7 (16.2-17.3) | 0.87 | 0.84-0.90 |
| Insufficiently Active | 21.2 (20.8-21.5) | 19.5 (18.9-20.1) | 0.92 | 0.89-0.95 |
| Inactive | 26.9 (26.5-27.3) | 29.9 (29.3-30.6) | 1.11 | 1.09-1.14 |

**Minimal Aerobic (2-level)**

| | Metropolitan, % (95% CI) (ref) | Non-Metropolitan, % (95% CI) | AOR | 95% CI |
|---|---|---|---|---|
| Meet | 51.0 (50.6-51.4) | 47.0 (46.3-47.6) | 0.94 | 0.91-0.97 |

**Muscle Strengthening**

| | Metropolitan, % (95% CI) (ref) | Non-Metropolitan, % (95% CI) | AOR | 95% CI |
|---|---|---|---|---|
| Meet | 36.1 (35.8-36.5) | 32.2 (31.5-32.8) | 0.92 | 0.89-0.96 |

**Combined Aerobic and Muscle Strengthening**

| | Metropolitan, % (95% CI) (ref) | Non-Metropolitan, % (95% CI) | RRR | 95% CI |
|---|---|---|---|---|
| Combined | 23.9 (23.5-24.2) | 22.6 (22.0-23.2) | 0.95 | 0.92-0.98 |
| Aerobic Only | 28.3 (27.9-28.6) | 27.9 (27.3-28.6) | 0.99 | 0.96-1.01 |
| Muscle Strengthening Only | 12.6 (12.3-12.8) | 12.3 (11.8-12.8) | 0.98 | 0.94-1.02 |
| Neither | 35.3 (34.9-35.8) | 37.2 (36.5-37.9) | 1.05 | 1.03-1.08 |

\*Adjusted for age, region, sex, race/ethnicity, education, income, month of survey response, self-reported health, and body mass index; Predicted Marginal Proportion for 4-level variables

Note: PAG: physical activity guideline; MVPA: moderate-to-vigorous intensity physical activity; AOR: Adjusted Odds Ratio; RRR: Relative Risk Ratio

Metropolitan = residing in a county within a Metropolitan Statistical Area; Non-Metropolitan = residing in a Micropolitan or Noncore county (outside of a Metropolitan Statistical Area), per 2013 National Center for Health Statistics (NCHS) Urban/Rural Classification Scheme

High Aerobic 4-level: Highly Active = 300 or more MVPA minutes per week; Active = 150–299 MVPA minutes per week; Insufficient = 1–149 MVPA minutes per week; Inactive = no leisure time MVPA

Minimal Aerobic 2-level: Meet = 150 or more MVPA minutes per week; Did Not Meet = 0–149 MVPA minutes per week

Muscle Strengthening: Meet = 2 or more days per week; Did Not Meet = 0–1 days per week

Combined Aerobic and Muscle Strengthening = cross tabulation of Aerobic (2-level) and Muscle Strengthening PAGs

### Prevalence of meeting the combined PAGs by LTPA category and metropolitan status

Displayed in Table 4 are the prevalence rates of meeting the combined PAGs by PA category compared by metropolitan residential status. Individuals who engaged in running/jogging, weightlifting, conditioning, other PA, and dancing had the highest rates of meeting the combined PAGs (all ≥ 44.9%). The lowest rates of meeting the combined PAGs regardless of residential status were achieved by those who engaged in walking, lawn/garden, fishing/hunting, and winter activities (all ≤ 32.4%). The highest rates of not meeting either PAG were witnessed in those who engaged in walking (22.7% in metropolitan and 23.2% in non-metropolitan residents).

There were significant differences in meeting the combined PAGs by metropolitan status for only 2 categories of activities: weightlifting and other PA. Among the respondents who engaged in weightlifting (10.7%), non-metropolitan residents (7.3%; 95% CI = 6.1-8.4%) had lower prevalence of not meeting either PAG than metropolitan residents (9.3%; 95% CI = 8.6-10.0%). Among the respondents who engaged in other PA (9.3%), non-metropolitan residents had greater prevalence of meeting only the aerobic PAG (36.9%; 95% CI = 35.0-38.9%) than metropolitan residents (31.8%; 95% CI = 30.6-33.0%).

### Discussion

The most prevalent types of LTPA in these analyses of 2019 BRFSS data were similar to those of US adults in the 2011 BRFSS data [8] and older NHANES data [17,18]. That is, walking continues to be the most common LTPA at roughly

**Table 4. Age-Adjusted Prevalence of Meeting the Combined-, Aerobic Only, Muscle Strengthening Only, and Neither Physical Activity Guideline in Leisure-Time by Physical Activity Category and Metropolitan Residential Status – 2019 BRFSS.**

| Physical Activity Category | Residence | Combined<br>% (95% CI) | Aerobic<br>% (95% CI) | Muscle Strengthening<br>% (95% CI) | Neither<br>% (95% CI) |
|---|---|---|---|---|---|
| Conditioning | Metropolitan | 44.9 (44.0-45.7) | 16.7 (16.1-17.3) | 26.0 (25.3-26.8) | 12.4 (11.8-12.9) |
| | Non-Metropolitan | 46.8 (45.1-48.6) | 15.4 (14.3-16.5) | 26.2 (24.8-27.7) | 11.6 (10.5-12.6) |
| Dance | Metropolitan | 50.9 (48.6-53.2) | 37.7 (35.5-39.9) | 4.0 (3.0-5.0) | 7.5 (6.4-8.6) |
| | Non-Metropolitan | 53.8 (49.4-58.2) | 33.8 (29.7-38.0) | 3.3 (1.8-4.7) | 9.1 (6.5-11.7) |
| Fishing/Hunting | Metropolitan | 32.4 (26.1-38.7) | 58.1 (51.5-64.7) | 0.7 (0.0-1.5) | 8.8 (5.2-12.3) |
| | Non-Metropolitan | 29.7 (23.3-36.2) | 60.9 (54.4-67.5) | 2.0 (0.1-3.9) | 4.3 (4.5-10.2) |
| Lawn/Garden | Metropolitan | 24.4 (22.9-25.9) | 59.6 (58.0-61.2) | 2.4 (2.0-2.9) | 13.6 (12.5-14.7) |
| | Non-Metropolitan | 25.3 (23.0-27.5) | 60.2 (57.8-62.7) | 2.5 (1.9-3.1) | 12.0 (10.5-13.6) |
| Running/Jogging | Metropolitan | 46.8 (44.8-48.8) | 29.4 (27.6-31.3) | 12.4 (11.7-13.2) | 11.4 (10.6-12.2) |
| | Non-Metropolitan | 47.4 (44.5-50.4) | 28.9 (26.3-31.6) | 13.5 (11.8-15.2) | 10.2 (8.6-11.7) |
| Sports | Metropolitan | 38.1 (36.7-39.5) | 46.4 (45.0-47.8) | 4.8 (4.2-5.4) | 10.7 (9.6-11.8) |
| | Non-Metropolitan | 35.1 (32.3-37.9) | 48.7 (45.9-51.5) | 4.7 (3.7-5.7) | 11.5 (9.7-13.4) |
| Walking | Metropolitan | 26.8 (26.3-27.3) | 43.5 (43.0-44.1) | 6.9 (6.6-7.2) | 22.7 (22.2-23.2) |
| | Non-Metropolitan | 26.2 (25.3-27.1) | 44.3 (43.3-45.3) | 6.3 (5.9-6.7) | 23.2 (22.4-24.1) |
| Water Activities | Metropolitan | 34.8 (32.8-36.9) | 47.4 (45.2-49.6) | 6.2 (4.9-7.5) | 11.6 (10.2-12.9) |
| | Non-Metropolitan | 30.7 (27.0-34.4) | 49.8 (45.8-53.8) | 5.2 (3.6-6.9) | 14.2 (11.1-17.4) |
| Winter Activities | Metropolitan | 31.4 (27.3-35.5) | 50.2 (45.7-54.8) | 5.3 (3.3-7.3) | 13.1 (10.3-15.8) |
| | Non-Metropolitan | 28.0 (22.9-33.0) | 52.0 (45.8-58.2) | 4.5 (1.9-7.1) | 15.5 (10.4-20.6) |
| Weightlifting | Metropolitan | 47.7 (46.6-48.9) | 6.2 (5.6-6.8) | 36.8 (35.7-38.0) | *9.3 (8.6-10.0)* |
| | Non-Metropolitan | 50.7 (48.3-53.1) | 5.1 (4.0-6.2) | 36.9 (34.6-39.2) | *7.3 (6.1-8.4)* |
| Other Physical Activity | Metropolitan | 50.1 (48.8-51.5) | *31.8 (30.6-33.0)* | 8.9 (8.1-9.7) | 9.2 (8.4-9.9) |
| | Non-Metropolitan | 47.0 (44.9-49.2) | *36.9 (35.0-38.9)* | 7.3 (6.3-8.4) | 8.7 (7.5-9.9) |

NOTE: The prevalence rates include only people who participated in a physical activity within that category (i.e., inactive respondents were not included). Significant differences between metropolitan and non-metropolitan residents are noted in bold and italicized text.

45% participation, with a wide margin between walking and the next most common groups of activities: running/jogging, weightlifting, conditioning exercises, and lawn/garden activities. Notably, our analyses indicate the LTPAs most prevalent among non-metropolitan (i.e., rural) residents are activities with a utilitarian purpose, including gardening, yard work, household activities, farm/ranch work, childcare, hunting, fishing, and snow shoveling. Except for childcare and household activities, these are also seasonal activities dependent on the natural environment and weather. Conversely, the LTPAs less prevalent among rural residents are higher intensity activities with a health and fitness focus (e.g., running, conditioning exercises, sports, weightlifting) that are reliant on the built environment (e.g., sports fields/courts, fitness centers) and additional resources (e.g., fitness instructors, sports league directors, parks and recreation staff) that may be limited or lacking in rural areas of the US [28,29].

Differences in the type and amount of LTPA engaged in by rural adults relative to urban adults may be a result of several sociocultural factors specific to rural areas of the US that constrain how leisure time is used and the availability of leisure time for PA. First, rural culture values adults' use of leisure time for additional manual labor or purposeful activities (e.g., farming, household maintenance) but not PA for the purpose of improving health [10]. Second, rural workers are more likely to be employed in occupations requiring manual labor and/or non-traditional work hours [30]. Consequently, survey research among samples of rural US adults have found tiredness as a common barrier to PA among rural men [31] and occupational PA as the primary PA domain linked to meeting PAGs with recreational PA (i.e., LTPA) being quite low

[32]. However, recent research suggests that high amounts of occupational PA are negatively associated with cardiovascular health [33,34], highlighting the need for greater emphasis on increasing LTPA. Third, the available time for leisure may also be constrained by higher rates of poverty [35], a point supported by research in urban areas showing that the provision of a guaranteed income alleviates time scarcity, anxiety, and poor health associated with income scarcity [36]. Finally, family and caregiving obligations are additional barriers for rural adults, especially women [37], further constraining the availability of leisure time for PA. These mechanisms are worth exploring in future descriptive and/or intervention research.

Our findings that rural (i.e., non-metropolitan) adults had a significantly higher prevalence of meeting the high aerobic PAG is novel, though minimally significant, and worth reassessing in 2023 BRFSS and future analyses of surveillance data. If replicated, it will be important to identify what LTPAs the high active group engages in and whether that group also meets the muscle-strengthening PAG. Consistent with extant literature, rural adults are more likely to be sedentary and less likely to meet aerobic, muscle strengthening, and combined PAGs [4,7]. Additionally, with the increased emphasis to "move more and sit less" in the *2018 Physical Activity Guidelines for Americans* [1], it is worth highlighting that urban (i.e., metropolitan) adults have a higher prevalence of being insufficiently active (1–149 minutes MVPA/week) and meeting the minimal activity threshold (150–299 minutes MVPA/week). Exact PAG prevalence rates should be compared with caution to previous research using pre-2019 NHIS data because of differences in analytic approaches and survey prompts (i.e., NHIS asked respondents to report only activities engaged in "at least 10 minutes at a time," whereas BRFSS excludes activities of <10 minutes in duration when calculating PA minutes and PAGs) [23,38].

Importantly, these analyses provide insight into prevalence of meeting PAGs by activity type. Unlike previous analyses [8], we included weightlifting and "other" PA in our study. Regardless of metropolitan status, roughly half of the respondents who engaged in weightlifting and other activities met the combined PAG. Conversely, walking stands out as the activity type with one of the lowest rates of meeting the combined PAGs and the highest rate of not meeting either PAG regardless of metropolitan status. It is discouraging that the activity that is by far the most popular is associated with such high rates of not meeting either PAG. Though novel, our findings are similar to research using NHANES 2003−04 and 2005−06 data collection cycles demonstrating that individuals who only engaged in walking had lower prevalence of meeting the aerobic PAG than those who engaged in 2 or more activities for all but the least active cluster of respondents [17,18]. Future research could add to our understanding by creating combinations of the 2 most frequent activities engaged in by BRFSS respondents (e.g., walking and weightlifting) and their association with PAGs to identify the most beneficial activity combinations. Changes in design of the BRFSS questionnaire to include more than the 2 most frequent activities engaged in, as was done in NHANES 2003−04 and 2005−06 cycles, may provide a more detailed picture of the associations among activities and PAGs, but should be considered in tandem with participant burden.

Overall, the activities more popular in metropolitan areas (i.e., running/jogging, weightlifting, conditioning, sports, dance) were associated with higher overall rates of meeting the combined PAGs than the activities more popular in non-metropolitan areas (i.e., lawn/garden, fishing/hunting, winter activities). This difference in activity preference may help explain the rural-urban disparities in meeting PAGs. There are numerous implications of these findings for rural PA promotion and link to the FITT principle of exercise prescription. First, the activity types associated with the highest prevalence of meeting the combined PAGs but lower prevalence among rural respondents (i.e., conditioning, dance, weightlifting, running) could be encouraged in non-metropolitan areas. This encouragement could meet resistance, however, for cultural, physical access, and/or demographic reasons (i.e., rural populations tend to be older and have higher rates of disability and chronic conditions) [39,40]. Dance or conditioning activities of cultural importance by geography or age could be considered when promoting PA in rural areas; activities for people of varying abilities due to age, injury, or existing chronic conditions could also be considered. The importance of promoting culturally relevant PA has been noted, for example, by authors from Hawaii who have used Hula for PA promotion/disease prevention [13] and begun surveilling Hula (and paddling) prevalence on the Hawaii BRFSS [12].

Second, as suggested by Burnet and colleagues [16], fun and beliefs (e.g., self-efficacy) should be considered in addition to FITT. That is, a fun, accomplishable activity may increase the likelihood of the uptake of an exercise prescription [16]. At the population level, this could be reflected by encouraging fun to increase engagement in the activities less popular in rural areas but associated with meeting PAGs. For example, engaging in activities with groups, such as a weightlifting or dance class or a running group, could help make the beneficial activities more fun and further improve health by strengthening social ties. Additionally, group activity could be used to increase the amount and types of PA when engaging in LTPAs of greater prevalence but lower association with meeting the combined PAGs. For example, walking groups are recommended in the Guide to Community Preventive Services [9] and have been shown to increase levels of PA, fitness, social support and connections, including with rural women [37,41,42]. These groups could be used with rural adult residents to increase aerobic PA and introduce them to muscle-strengthening activities in order to meet the combined PAG.

Third, engaging in LTPA for a more tangible or meaningful purpose than preventing or treating disease may be worth exploring to increase LTPA in rural areas. That is, linking LTPA engagement to benefits in the hunting, fishing, and child-care activities more prevalent in non-metropolitan areas may be effective in rural areas. For example, deer hunting is associated with large heart rate responses, heart attacks, and cardiovascular complications [43,44]. Thus, a multi-component intervention targeting hunters could be implemented at the locations or websites where hunting licenses are sold to integrate messages about PA (e.g., "Struggling to drag that deer out of the woods? Be more active year-round and drag that deer with ease.") and/or a PA screening tool that could inform individuals at risk to seek a health care provider's advice before going hunting [45].

The findings presented herein are novel, but worth viewing considering multiple limitations. First, rurality was defined by county of residence in the BRFSS dataset as within- or outside of a metropolitan statistical area. While this definition is common in the literature, it has at least 2 limitations – it lacks details about each respondent's proximity to a metropolitan area and it lacks the assessment of the location of LTPA. To emphasize, an individual could live near a metropolitan county or commute to a metropolitan area for work and thus have easy access to the resources of that metropolitan area for LTPA. These are complications, hopefully attenuated by the large sample and population weighting employed in the analyses. Second, the 2 most common LTPAs in the previous month were assessed in the BRFSS. Though previous research suggests this encompasses roughly 80% of population preferences [18], it does not capture all LTPAs that each respondent participates in. Third, recall bias may impact the accurate self-reporting of activity types and/or PA duration by respondents. Lastly, the BRFSS assesses LTPA only, not PA engaged in for transportation or occupational purposes. Though the PAGs are based on LTPA, it does not give a complete assessment of PA across all domains of activity.

## Conclusions

Walking continues to be the most popular LTPA among US adults. However, those that walk as one of their two most frequent LTPAs have the highest rate of not meeting either PAG. Further, the activities preferred by non-metropolitan US adults tended to be more utilitarian and dependent on the natural environment whereas higher intensity, health-focused activities reliant on the built environment were more often preferred by metropolitan residing US adults. These findings have several implications for future research and practice. Future analyses could explore combinations of LTPAs to see if certain pairs of activities (e.g., walking plus weightlifting) are associated with meeting the combined PAGs. In addition, replicating these analyses using 2023 BRFSS data is encouraged because patterns of LTPA shifted during the COVID-19 pandemic [46]. With respect to public health practice, population-level LTPA intervention efforts in rural areas could be impacted in divergent ways. That is, PA preferences could be *embraced and augmented* (e.g., walking is popular but need more types of activities to meet the combined PAGs) or *non-preferred activities could be promoted* because of their association with the combined PAGs (e.g., weightlifting, dancing) but in a culturally-, accessibility-, and demographically-adapted way (e.g., chair exercise for older adults; clogging or another dance endemic to a certain rural

geography). Regardless, these analyses enhance our understanding of the LTPA preferences and association with PAGs among US adults residing in rural and urban communities.

## Acknowledgments

The authors would like to acknowledge the Centers for Disease Control and Prevention (CDC), National Center for Chronic Disease Prevention and Health Promotion, Division of Nutrition, Physical Activity, and Obesity, including the Physical Activity and Health Branch, for their ongoing dedication to maintaining, updating, and openly sharing the BRFSS database and increasing physical activity in the US.

## Author contributions

**Conceptualization:** Christiaan Abildso, Eugene C. Fitzhugh, Alan M. Beck, Ashleigh Johnson, Dina L. Maruca, Stefanie M. Meyer, Cynthia K. Perry, Carissa R. Smock, Zachary Townsend, Lauren E. Jacobs, M. Renée Umstattd Meyer.

**Data curation:** Eugene C. Fitzhugh.

**Formal analysis:** Eugene C. Fitzhugh.

**Methodology:** Christiaan Abildso, Eugene C. Fitzhugh, Alan M. Beck, Ashleigh Johnson, Dina L. Maruca, Stefanie M. Meyer, Cynthia K. Perry, Carissa R. Smock, Zachary Townsend, M. Renée Umstattd Meyer.

**Project administration:** Christiaan Abildso.

**Supervision:** M. Renée Umstattd Meyer.

**Visualization:** Eugene C. Fitzhugh.

**Writing – original draft:** Christiaan Abildso, Eugene C. Fitzhugh.

**Writing – review & editing:** Alan M. Beck, Ashleigh Johnson, Dina L. Maruca, Stefanie M. Meyer, Cynthia K. Perry, Carissa R. Smock, Zachary Townsend, Lauren E. Jacobs, M. Renée Umstattd Meyer.

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
