## [Decision Letter · Decision Letter 0]

4 Jan 2026

Dear Dr. Abildso,

Thank you for submitting your manuscript to PLOS ONE. After careful consideration, we feel that it has merit but does not fully meet PLOS ONE’s publication criteria as it currently stands. Therefore, we invite you to submit a revised version of the manuscript that addresses the points raised during the review process.

We look forward to receiving your revised manuscript.

Kind regards,

Zulkarnain Jaafar

Academic Editor

PLOS One

**Journal Requirements:**

“I have read the journal's policy and the authors of this manuscript report receiving no financial support for this work or any competing interests. However, Dr. Johnson was supported by a Research Career Development Award from the National Heart, Lung, and Blood Institute (K01HL171860-01) during the period in which the research took place.”

**Additional Editor Comments:**

Dear Author, please make necessary corrections based on the comments provided by the reviewers.

Reviewers' comments:

Reviewer's Responses to Questions

**Comments to the Author**

1. Is the manuscript technically sound, and do the data support the conclusions?

Reviewer #1: Partly

Reviewer #2: Yes

2. Has the statistical analysis been performed appropriately and rigorously?

Reviewer #1: Yes

Reviewer #2: Yes

3. Have the authors made all data underlying the findings in their manuscript fully available?

Reviewer #1: Yes

Reviewer #2: Yes

4. Is the manuscript presented in an intelligible fashion and written in standard English?

Reviewer #1: Yes

Reviewer #2: Yes

Reviewer #1: I have carefully read the manuscript which shows a valuable contribution with a robust scientific approach. I suggest the following revisions be addressed before the manuscript is ready for publication.

Comment 1. In line 139 and 150, please clarify if PROC DESCRIPT and PROC RLOGIST are a procedure or something else.

Comment 2. Please clarify if the assumptions logistic regression were met.

comment 3. The manuscript lacks of a conclusion section, which is essential for summarizing main findings.

Reviewer #2: I appreciate the opportunity to review this manuscript. The article is well written and presents an interesting analysis of public epidemiological data. However, below are some comments aimed at improving certain aspects.

L24: Please consider using a less categorical statement and change “will only be realized” for something like “may be achieve if” or similar.

L26: Please remove “high aerobic” since it does not fit here while you are describing modes of exercise.

L91: Readers could be interested in further explanation about PA categories defined by Watson and colleagues; for example: what does “conditioning” include? Why did you decide to include “weightlifting” as an additional category instead of incorporating into “conditioning”? What are the differences between “sports”, “water activities” and “winter activities”? For example, if somebody practice skiing: Would it be included into “sports” or into “winter activities”?

L101, 109 and 114: Please avoid using PAG for section subtitles because you are not describing guidelines but respondents’ leisure-time physical activity (LTPA) participation. The same occurs during the text (L104: “for a dichotomous “minimal aerobic” PAG” should be rewritten as “for a dichotomous “minimal aerobic” LTPA”; L106: “the minimal aerobic PAG” should be rewritten as “the minimal aerobic LTPA”; etc.).

L107: I cannot understand why you name “high aerobic” for this 4-level categories… I consider that using “high” is confusing… Actually, in table 3 you don’t use “high” when describing results for this 4 aerobic level.

L126: Use a different abbreviation for “metropolitan statistical area” to avoid confusion with “muscle-strengthening activity”.

In addition to the limitation declare by authors, the limitation about the fact that it uses categories of physical activity that are not well suited to rural areas (such as Yoga or Pilates) and are carried out exclusively during leisure time should be further defined. Also, terminologically, it could refer more to physical exercise than to physical activity, since the physical activity inherent in the lifestyle derived from professional activities or domestic responsibilities has not been analyzed. In this sense, it could be that people living in rural areas who have been classified as “inactive” could have similar or even higher levels of physical activity if the activity practiced outside leisure time were analyzed. The authors should reflect this limitation and mention it in other sections of the manuscript to avoid biased conclusions. Additionally, as authors described, the method used to record physical activity in the original study lacks rigor, meaning that the data recorded may not reflect the social reality that it aims to analyze. Future studies should use validated methods for quantifying physical activity as well as include physical activity practiced outside of leisure time. Lastly, authors may consider mentioning as future study to analyze PA preferences to improve physical activity promotion campaigns.

**Do you want your identity to be public for this peer review?** For information about this choice, including consent withdrawal, please see our Privacy Policy

Reviewer #1: No

Reviewer #2: No

---

## [Author Response · Author response to Decision Letter 1]

23 Jan 2026

A file with all responses has been uploaded. It is also pasted below with Responses denoted with "R: " following each comment

PONE-D-25-17562: Response to Reviewers

Title: Adults’ Leisure-Time Physical Activity Preferences and Association with Physical Activity Guidelines by Metropolitan Status, United States, 2019

Thank you for the thorough review of the manuscript and for the opportunity to revise and resubmit. Please find a point by point summary of changes made to each of the reviewer comments in the following table. Revised manuscript files uploaded include one that utilizes track changes mode to denote the edits (“Revised Manuscript with Track Changes”) and another that does not utilize track changes mode (“Manuscript”).

In addition to the response to the reviewers we (1) updated the title page to fit the PLOS One format, (2) revised the title of Table 4 to more accurately represent what is in the table, and (3) noted in the cover letter that the funding received by any of the co-authors “does not alter our adherence to PLOS ONE policies on sharing data and materials”

====

Reviewer 1: I have carefully read the manuscript which shows a valuable contribution with a robust scientific approach. I suggest the following revisions be addressed before the manuscript is ready for publication

Comment 1. In line 139 and 150, please clarify if PROC DESCRIPT and PROC RLOGIST are a procedure or something else.

R: Thank you for this comment. These are the names of the procedures in the SAS software. We added “the SAS procedure…” to lines 139, 150, and 154 to clarify.

Comment 2. Please clarify if the assumptions logistic regression were met.

R: Yes, assumptions for all statistical procedures, including the RLOGIST and MULTILOG, were examined. Related to the regression models we specifically focused linearity of the logit and multicollinearity.

We include a brief statement in the first sentence of the statistical analyses that assumptions were also examined across the procedures.

Comment 3. The manuscript lacks of a conclusion section, which is essential for summarizing main findings.

R: Thank you for this comment. We revised the final paragraph of the Discussion section to be the Conclusion section which includes a brief summary of results and implications for research and practice.

Reviewer 2:

I appreciate the opportunity to review this manuscript. The article is well written and presents an interesting analysis of public epidemiological data. However, below are some comments aimed at improving certain aspects:

L24: Please consider using a less categorical statement and change “will only be realized” for something like “may be achieve if” or similar.

R: Thank you for this suggestion. We revised the first sentence to be more general about the importance of physical activity for public health.

L26: Please remove “high aerobic” since it does not fit here while you are describing modes of exercise.

R: Removed. Thank you.

L91: Readers could be interested in further explanation about PA categories defined by Watson and colleagues; for example: what does “conditioning” include? Why did you decide to include “weightlifting” as an additional category instead of incorporating into “conditioning”? What are the differences between “sports”, “water activities” and “winter activities”? For example, if somebody practice skiing: Would it be included into “sports” or into “winter activities”?

R: These are excellent questions that we discussed quite a bit as well. We revised the Physical Activity Measures section to reflect what is described below.

We used the Compendium of Physical Activities for the categorization of specific physical activities. The Compendium was created to address the need for consistency in assigning intensity levels of PA captured from questionnaires used in epidemiological studies, including the BRFSS. We wanted to maintain that consistency as well. *Please note, we cited the previous edition of the Compendium in error. We used the 3rd edition, published in 2024, when categorizing activities. That citation (#23) has been updated.

With respect to the differentiations between activities and the purpose of those activities, the Compendium organizes specific activities under major headings that are used to classify PA by its primary purpose or activity domain. However, the BRFSS reduces the list to 74 specific activities to simplify the coding process, often by eliminating the “purpose” and other activity specifics.

For example, the Compendium Winter Activities heading includes 9 different cross-country and downhill skiing codes based on speed, effort, and intensity levels. The BRFSS has only “snow skiing” as a coded activity type related to skiing.

One exception to that coding is weightlifting as the reviewer noted. We chose to include weightlifting and other anaerobic / muscle strengthening activities because our analyses included the muscle strengthening PAG as an outcome – unlike Watson et al which only focused on aerobic physical activities. We also chose to have “Weightlifting” as its own category rather than be listed under the “Conditioning” major heading, to maintain consistency with Watson et al.’s conditioning category. The Compendium does list Weightlifting under the Conditioning heading, as the reviewer indicates, but it is the only activity under the Conditioning heading that is coded by the BRFSS that is specifically focused on resistance training/muscle strengthening activity.

L101, 109 and 114: Please avoid using PAG for section subtitles because you are not describing guidelines but respondents’ leisure-time physical activity (LTPA) participation. The same occurs during the text (L104: “for a dichotomous “minimal aerobic” PAG” should be rewritten as “for a dichotomous “minimal aerobic” LTPA”; L106: “the minimal aerobic PAG” should be rewritten as “the minimal aerobic LTPA”; etc.).

R: Thank you for this suggestion. To clarify, we are using the BRFSS variables that categorize respondents as meeting the physical activity guidelines or not. They are not just based on participating in any activity or not participating. We changed the subtitles as suggested but did not make changes to the text because the variables are based on meeting or not meeting the PAGs. We also moved one sentence describing how the BRFSS categorizes aerobic PA, improving the flow, and revised “PA” to be “LTPA” in a few instances to be consistent with the rest of the text.

L107: I cannot understand why you name “high aerobic” for this 4-level categories… I consider that using “high” is confusing… Actually, in table 3 you don’t use “high” when describing results for this 4 aerobic level.

R: *We revised Table 3 as suggested and to be consistent with the text.

NB: The labeling of “high aerobic” comes from the Physical Activity Guidelines for Americans (2nd Edition) and is consistent with previous research (Whitfield, G. P., Hyde, E. T., & Carlson, S. A. (2021). Participation in Leisure-Time Aerobic Physical Activity Among Adults, National Health Interview Survey, 1998–2018. Journal of Physical Activity and Health, 18(S1), S25-S36. https://doi.org/10.1123/jpah.2021-0014.)

L126: Use a different abbreviation for “metropolitan statistical area” to avoid confusion with “muscle-strengthening activity”.

R: Thank you for catching this! MSA is more commonly used for metropolitan statistical area. We used it in that way. Thus, we wrote out “muscle-strengthening activity” rather than using an acronym for that term.

Comment: In addition to the limitation declare by authors, the limitation about the fact that it uses categories of physical activity that are not well suited to rural areas (such as Yoga or Pilates) and are carried out exclusively during leisure time should be further defined. Also, terminologically, it could refer more to physical exercise than to physical activity, since the physical activity inherent in the lifestyle derived from professional activities or domestic responsibilities has not been analyzed. In this sense, it could be that people living in rural areas who have been classified as “inactive” could have similar or even higher levels of physical activity if the activity practiced outside leisure time were analyzed. The authors should reflect this limitation and mention it in other sections of the manuscript to avoid biased conclusions. Additionally, as authors described, the method used to record physical activity in the original study lacks rigor, meaning that the data recorded may not reflect the social reality that it aims to analyze. Future studies should use validated methods for quantifying physical activity as well as include physical activity practiced outside of leisure time. Lastly, authors may consider mentioning as future study to analyze PA preferences to improve physical activity promotion campaigns.

R: Thank you for these thoughtful comments. To address these comments, the Physical Activity Measures section of the Methods section was revised.

With regard to the categories of PA, our goal was to expand the seminal work of Watson et al (2015) by adding the rural/urban comparison and assessing the relationship with meeting PAGs. We did not adjust the categories, with the exception of weightlifting / conditioning noted in a previous comment because we wanted to be able to compare with that study and because those categories are based on the Compendium of Physical Activities, another seminal work.

We concur that the focus on leisure-time physical activity is a limitation of the BRFSS data. This was noted as a limitation in the original submission (“Lastly, the BRFSS assesses LTPA only, not PA engaged in for transportation or occupational purposes. Though the PAGs are based on LTPA, it does not give a complete assessment of PA across all domains of activity,”) just prior to the final paragraph (Conclusion) of the manuscript.

We believe the comment about leisure-time also addresses the reviewer’s suggestion about using the term “physical exercise.” The BRFSS uses a validated measure of PA that begins by asking participants “During the past month, other than your regular job, did you participate in any physical activities or exercises such as running, calisthenics, golf, gardening, or walking for exercise?” This instrument includes exercise/fitness and household/domestic activities performed “outside of work hours.”

---

## [Decision Letter · Decision Letter 1]

2 Mar 2026

Adults’ leisure-time physical activity preferences and association with physical activity guidelines by metropolitan status, United States, 2019

PONE-D-25-17562R1

Dear Dr. Abildso,

We’re pleased to inform you that your manuscript has been judged scientifically suitable for publication and will be formally accepted for publication once it meets all outstanding technical requirements.

Kind regards,

Zulkarnain Jaafar

Academic Editor

PLOS One

Additional Editor Comments (optional):

Reviewers' comments:

Reviewer's Responses to Questions

**Comments to the Author**

Reviewer #1: All comments have been addressed

Reviewer #2: All comments have been addressed

2. Is the manuscript technically sound, and do the data support the conclusions?

Reviewer #1: Yes

Reviewer #2: Yes

3. Has the statistical analysis been performed appropriately and rigorously?

Reviewer #1: Yes

Reviewer #2: Yes

4. Have the authors made all data underlying the findings in their manuscript fully available?

Reviewer #1: Yes

Reviewer #2: Yes

5. Is the manuscript presented in an intelligible fashion and written in standard English?

Reviewer #1: Yes

Reviewer #2: Yes

Reviewer #1: (No Response)

Reviewer #2: The authors have satisfactorily addressed the suggestions made or, where this has not been possible, have provided adequate justification for their responses.

**Do you want your identity to be public for this peer review?** For information about this choice, including consent withdrawal, please see our Privacy Policy

Reviewer #1: No

Reviewer #2: No

---

## [Editor Report · Acceptance letter]

PONE-D-25-17562R1

PLOS One

Dear Dr. Abildso,

I'm pleased to inform you that your manuscript has been deemed suitable for publication in PLOS One. Congratulations! Your manuscript is now being handed over to our production team.

Kind regards,

on behalf of

Dr. Zulkarnain Jaafar

Academic Editor

PLOS One